# Hepatic insulin synthesis increases in rat models of diabetes mellitus type 1 and 2 differently

**Musa Abidov**[1], **Ksenia Sokolova**[2], **Irina Danilova**[2], **Madina Baykenova**[3], **Irina Gette**[2], **Elena Mychlynina**[2], **Burcin Aydin Ozgur**[4,5]*, **Ali Osman Gurol**[6,7], **M. Temel Yilmaz**[8]

**1** Institute of Immunopathology and Preventive Medicine, Ljubljana, Slovenia, **2** Institute of Immunology and Physiology, Ural Branch of the Russian Academy of Sciences, Yekaterinburg, Russian Federation, **3** Kostanay Oblast Tuberculosis Dispensary, Kostanay, Republic of Kazakhstan, **4** Department of Medical Biology and Genetics, Faculty of Medicine, Demiroglu Bilim University, Istanbul, Turkey, **5** Diabetes Application and Research Center, Demiroglu Bilim University, Istanbul, Turkey, **6** Department of Immunology, Aziz Sancar Institute of Experimental Medicine, Istanbul University, Istanbul, Turkey, **7** Diabetes Application and Research Center, Istanbul University, Istanbul, Turkey, **8** International Diabetes Center, Acibadem University, Istanbul, Turkey

\* burcin-aydinozgur@hotmail.com

**Data Availability Statement:** All relevant data are within the paper and its Supporting Information files.

## Abstract

Insulin-positive (+) cells (IPCs), detected in multiple organs, are of great interest as a probable alternative to ameliorate pancreatic beta-cells dysfunction and insulin deficiency in diabetes. Liver is a potential source of IPCs due to it common embryological origin with pancreas. We previously demonstrated the presence of IPCs in the liver of healthy and diabetic rats, but detailed description and analysis of the factors, which potentially can induced ectopic hepatic expression of insulin in type 1 (T1D) and type 2 diabetes (T2D), were not performed. In present study we evaluate mass of hepatic IPCs in the rat models of T1D and T2D and discuss factors, which may stimulate it generation: glycaemia, organ injury, involving of hepatic stem/progenitor cell compartment, expression of transcription factors and inflammation. Quantity of IPCs in the liver was up by 1.7-fold in rats with T1D and 10-fold in T2D compared to non-diabetic (ND) rats. We concluded that ectopic hepatic expression of insulin gene is activated by combined action of a number of factors, with inflammation playing a decision role.

## Introduction

Currently diabetes mellitus (DM) is one of the highly prevalent socially significant chronic diseases, with a high disability of patients. Prevalence of diabetes is growing every year, and it is classified as one of the leading causes of high mortality in developed countries [1]. Diabetes mellitus is a group of metabolic diseases characterized by permanent hyperglycemia, resulting from absolute or relative insulin deficiency, insulin action or both [2]. Two main forms of diabetes mellitus are type 1 (T1D) and type 2 (T2D). T1D is an autoimmune and/or idiopathic disease, characterized by catastrophic loss of beta-cells and significant hyperglycemia [3], T2D is associated with

**Funding:** This research was carried out within the state assignment of the Institute Immunology and Physiology Ural Branch Russian Academy of Science (IIP UB RAS) 122020900136-4. The funders had no role in study design, data collection and analysis, decision to publish, or preparation of the manuscript. The work was performed using the equipment of the Shared Research Center of Scientific Equipment of the IIP UB RAS.

**Competing interests:** The authors have declared that no competing interests exist.

insulin resistance and medium hyperglycemia [4]. Loss and dysfunction of beta-cells are the main reasons of hyperglycemia and its associated complications in both types of DM [5].

Many investigators had reported that pancreatic islet beta-cells are not the single source of insulin in the body. Cells in different tissues beside pancreatic islets are also capable to synthesize insulin; insulin-positive cells (IPCs) were found in the exocrine pancreas, liver, thymus, brain, bone marrow, spleen and adipose tissue [6–8]. Extra-islet and extra-pancreatic IPCs are the spotlight of researchers for at least two reasons: first, they are additional sources of insulin and probably are able to alleviate hyperglycemia and prevent complications of diabetes. Secondly, ectopic insulin expression in functional mature cell in different tissues is of fundamental theoretical interest and expand the understanding of tissue adaptation in pathology condition.

What stimulates ectopic insulin expression in non-pancreatic tissues? Research data testify that mature cells may be reprogrammed and change their phenotype [9, 10]. Liver originated from the same region of the endoderm that pancreas and hepatic cells are able to transdifferentiate into pancreatic cells and vice versa [11–15]. In liver IPCs are detected as well in healthy [16], as in diabetic animals and/or under glucose load [7, 8, 16–19], and researchers consider that the conversion of hepatocytes into insulin-producing cells could be the basis of a new type of therapy for diabetes. We aimed to identify factors, impact of generation of IPCs in liver in the rat models of T1D and T2D in vivo.

Glucose is the main physiological regulator of insulin gene expression. Produced by glucose injections hyperglycemia in non-diabetic or diabetic mice led to the appearance of proinsulin- and insulin-positive cells in liver, adipose tissue and bone marrow within 3 days [7]. However, hyperglycemia is not the only factor influencing the emerging of IPCs in liver.

Maturation of beta-cells and synthesis of insulin are regulated by sequential expression of transcription factors. The most essential factors for development of beta-cells are pancreas and duodenal homeobox protein 1 (Pdx1), Neurogenin-3 (Ngn3) and V-maf musculoaponeurotic fibrosarcoma oncogene homolog A (MafA) [20]. Pdx1 is a key regulator for generation, maturation and functioning of pancreatic beta-cells in adult organisms. Pdx1-engineered embryonic stem cells robustly differentiate into glucose responsive insulin-producing cells [21]. Ectopic expression of Pdx1 induces sustained changes in hepatocytes, including activation of many specific pancreatic transcription factors and leads to production and secretion of insulin, which ameliorated hyperglycemia in STZ-induced diabetic mice [22]. Mice deficient for the Ngn3 fails to the develop of endocrine cells in intestine and pancreas and shows partial endocrine differentiation in stomach [23]. MafA is associated with maintaining beta-cell function as a regulator of genes involved in metabolism-secretion coupling, proinsulin processing and glucagon-like peptide-1 signaling [24]. A lot of experimental data show, that transcription factors Pdx1, Ngn3 and MafA are able to reprogram hepatocytes into glucose-responsive insulin-producing cells [25–27].

Due to the high phenotypic and functional heterogeneity of liver cells and peculiarities of liver zonal regeneration [28], the predisposition of hepatic cells to activate insulin-producing function in response to ectopic expression of Pdx1 should be affected by liver zonation. Cells located in different areas of liver have different ability to transdifferentiate and exhibit different degrees of the cellular plasticity [15]. Organ-specific stem cells possess plasticity that permit differentiation along new lineages. Hepatic oval cells are thought to have the potential to transdifferentiate into IPCs [29–32]. It is known that proliferation of oval cells has positive correlation with liver damage degree [33].

Both T1D and T2D are chronically inflammatory diseases, so all events in the body are developed in the pro-inflammatory background, which is known as associated with metabolic process. Inflammation is able to induce cell transition during regeneration [34]. Early we

revealed, that at the 30 days of experimental T2D the mass of solitary IPCs in pancreatic acinar increases [35].

In this study we evaluate the mass of hepatic IPCs in rat models of T1D and T2D with the equal hyperglycemia, but different expression of pancreatic transcription factors, number of oval cells and severity of inflammation and tissue damage.

## Materials and methods

### Animals

Experiments were performed on 30 12-weeks old male Wistar rats of weight 303.0 ± 25.3 g breeding in the animal facility of the Institute of Immunology and Physiology, Ural Branch Russian Academy of Science (Yekaterinburg, Russian Federation) and showing no disease symptoms. Rats were housed 5 per polypropylene cage and kept under equal laboratory conditions (12h light/12h dark cle with temperature 22 ± 2˚C above zero). Animals were fed according to customary schedule with soy protein-free extruded rodent diets (2020X Teklad, Envigo, Huntingdon, United Kingdom) and allowed free access to filtered tap water.

**Institutional review board statement.** All experimental procedures involving animals were in compliance with the applicable laws and regulations as well as the principles expressed in Directive 2010/63/EU of the European Parliament and of the Council of 22 September 2010 on protection of animals used for scientific purposes. The use of animals was approved by Ethical Committee of Institute of Immunology and Physiology of Russian Academy of Science (protocol number 04/19 from 18 Dec 2019).

### Induction of experimental diabetes and grouping

T1D was modeled by 3 i.p. injections of alloxan monohydrate (DIA-M, Russian Federation, at. No. U2244113.0050) dissolved in 0.85% NaCl, with total dose of 170 mg/kg b.w. [36]. T2D was induced by a single i.p. injection of 65 mg/kg b.w. streptozotocin (Stz) (Sigma-Aldrich, St Louis, MO, United States, at. No. S0130), 15 min after injection of 110 b.w. of nicotinamide (NA) (Sigma-Aldrich, St Louis, MO, United States, at. No. N3376-100G) [37]. The dry powder of Stz was dissolved in 0.1M citrate buffer (pH 4.5), while NA was dissolved in distilled water. All animals received freshly prepared solutions of drugs after 16 h of fasting.

Fasting blood glucose (FBG) level was measured 72 h after injection of diabetic agents and rats with FBG level of $\geq$ 7.1 mmol/L were selected as successfully prepared diabetic models and used in the study.

Animals were randomly divided into three groups, each group comprised ten rats (n = 10). Group 1 –non-diabetic (ND) age-matched intact control rats, group 2 –rats with alloxan-induced T1D, group 3 –rats with Stz/NA-induced T2D. In 30 days after the first injections of diabetogenic agents the rats were deeply anesthetized with an i.m. injection of 0.1 mg/kg b.w. xylazine (Alfasan,Woerden, Netherlands) and euthanized by i.m. injection of 5 mg/kg b.w. Zoletil-100 (Virbac, Carros, France).

### Fasting blood glucose (FBG), glycosylated hemoglobin (HbA1c), and oral glucose tolerance test (OGTT)

Blood samples were collected before euthanasia from the rats via the tail vein and centrifuged at 500x g for 10 min at 4˚C above zero. The plasma supernatant was using for biochemical tests using a DU-800 spectrophotometer (Beckman Coulter Int. S.A., Switzerland) at specified wavelength according to manufacturers. FBG level was determined by the glucose oxidase method using diagnostic kit (Vektor-Best, Russian Federation). Level of HbA1c was measured

using kit for affinity chromatography (GLICOHEMOTEST, ELTA, Russian Federation). To evaluate glycemic control in animals with T2D OGTT was done at the 4th week of the experiment (n = 6). Assessment was based on FBG level at 0 min, and postprandial glucose (PG) level measured after the oral administration of glucose (1 g/kg b.w.) at 30, 60, and 120 min. Administration of glucose was conducted using oral dosing needles (VetTech Solutions Ltd., United Kingdom). The area under the curve (AUC) was calculated according to Sakaguchi et al. (2016) using the trapezoidal approximation of PG levels [38].

## Serum insulin level and HOMA-estimated insulin resistance

Serum insulin levels were measured using Rat Insulin ELISA kit (Thermo Fisher Scientific, Waltham, MA, United States). Insulin resistance was estimated by homeostasis model assessment of insulin resistance (HOMA IR), according to method described by Matthews et al. [39, 40]. It was computed, using formula:

HOMA IR = fasting insulin (μU/mL) × fasting glucose (mmol/L) / 22.5.

## Evaluation of hepatic parameters

Plasma levels of aspartate aminotransferase (AST), alkaline phosphatase (ALT), alanine aminotransferase (ALP) and total protein were determined using ready to use reagents kits (Vital diagnostics, Russian Federation).

## Evaluation of hematological parameters

The number of leukocytes (white blood cells, WBC) and lymphocytes in heparinized blood samples was estimated using an automated hematology analyzer Celly 70 (Biocode Hycel, France). Plasma level of total protein was measured using standard reagent kit (Vital Diagnostics, Russian Federation).

## Histological and immunohistochemical study

Samples of liver tissue were excised and after 24 h fixation in 10% formalin were embedded in paraffin for histological assessment. Tissues were processed using tissue processor Leica TP 1020. Paraffin blocks were made on paraffin embedding station Leica EG 1160. 4–5 μm sections of liver tissue were cut, using sliding microtome Leica SM 2000R. The periodic acid-Schiff staining (PAS) was performed using standard kit for PAS 07-010/S (BioVitrum, Russian Federation).

Liver paraffin sections were studied using immunohistochemistry (IHC) and fluorescent IHC. Insulin+, F4/80+ cells (macrophages), white blood cells (CD45+ cells), lymphocytes (CD3+ cells), KRT19+ and OV6+ cells (oval cells) were detected by IHC, while Pdx1+, MafA + and Ngn3+ cells were detected using fluorescent IHC. Tissues were labeled with primary antibodies (Table 1) overnight, followed by incubation with secondary antibodies for 1 hour.

The immunohistochemical procedure was performed using the avidin-biotin peroxidase complex method. Staining was performed according to the manufacturer's standard protocols. To check the protocol and exclude nonspecific staining, negative and positive controls were set. Sections of the pancreas of non-diabetic rats were used as positive control [41–43]. As the negative control we used substitution of isotype-specific immunoglobulins at same protein concentration as primary antibody [42].

**Table 1. List of antibodies used for immunohistochemical and immunofluorescence studies.**

| Detected antigen | Primary antibodies: reference, supplier, dilution | Secondary antibodies: reference, supplier, dilution |
|---|---|---|
| Insulin | Anti-Insulin/Proinsulin: clone INS04+INS05, MA5-12042, Invitrogen, United States, 1:200 | Biotin Goat anti-Mouse Ig (Multiple Absorption), BD Pharmingen, United States, 1:500 |
| Pdx1 | Anti-PDX1: ab 227586, Abcam, United States, 1:200 | Goat anti-Rabbit IgG (H+L) + Texas Red, Thermo Fisher Scientific, United States, 1:100 |
| MafA | Anti-MafA: orb11013, Biorbyt, United States, 1:200 | Goat anti-Rabbit IgG (H+L) + Texas Red, Thermo Fisher Scientific, United States, 1:100 |
| Ngn3 | Anti-Neurogenin 3: 11127, Biorbyt, United States, 1:200 | Goat anti-Rabbit IgG (H+L) + Texas Red, Thermo Fisher Scientific, United States, 1:100 |
| OV6 | Anti-OV6 (R&D Systems, United States, MAB2020), 1:50 | Biotin Goat anti-Mouse Ig (Multiple Absorption), BD Pharmingen, United States, 1:50 |
| KR19 | Anti-KR19 (CUSABIO, United States, CSB-PA100243), 1:100 | Goat anti-Rabbit IgG (H+L), Thermo Fisher Scientific, United States, 1:500 |
| F4/80 | Anti-F4/80 Polyclonal Antibody, PA5-21399, Thermo Fisher Scientific, United States, 1:200 | Biotin Goat Anti-Rabbit IgG, Thermo Fisher, Scientific, United States, 1:50 |
| CD3 | Purified Mouse Anti-Rat CD3 clone G 4.18, BD Pharmingentm, United States, 1:30 | Biotin Goat Anti-Mouse Ig (Multiple Adsorption) (BD Biosciences, United States), 1:50 |
| CD45 | Purified Mouse Anti-Rat CD45 clone -1 (BD Pharmingentm, United States), 1:30 | Biotin Goat Anti-Mouse Ig (Multiple Adsorption) (BD Biosciences, United States), 1:50 |

## Morphometric analysis

Number of insulin+, F4/80+ cells (macrophages), white blood cells (CD45+ cells), lymphocytes (CD3+ cells), KRT19+ and OV6+ cells (oval cells) and sinusoidal cells were counted under the light microscope (Leica DM 2500) using ×400 and 1000 magnification with Las 4.9 Software (Leica Microsystems GmbH). Immunofluorescence images were obtained using confocal laser scanning microscope Carl Zeiss LSM710 at magnification ×400 with ZEN 2.0 software (Carl ZEISS, Germany). All morphometric evaluations were based on 20 microscopy fields per tissue section of each rat and were converted to the 1 mm$^2$ of liver parenchyma (N/mm$^2$).

## Statistical analysis

Statistical analysis was performed using Statistica 6.0 (StatSoft, Inc). Data were presented as mean ± standard error of the mean (SEM). SEM was calculated by dividing the standard deviation (SD) of the sample by the square root of the sample size (n) (SEM = SD/$\sqrt{}$n). Further data were analyzed by nonparametric Mann-Whitney U-test and Kruskal-Wallis test. P < 0.05 was regarded as statistically significant.

## Results

### Fasting blood glucose (FBG), glycosylated hemoglobin (HbA1c), glucose tolerance, serum insulin concentration and HOMA-estimated insulin resistance

As illustrated in Fig 1A and 1B, FBG and HbA1 levels in rats with T1D and T2D were approximately similar. Since glucose is identified as the main regulator establishing generation of IPCs in liver by many investigators, it was interesting for us to compare the quantity of IPCs in liver in the models of T1D and T2D with the similar glycaemia.

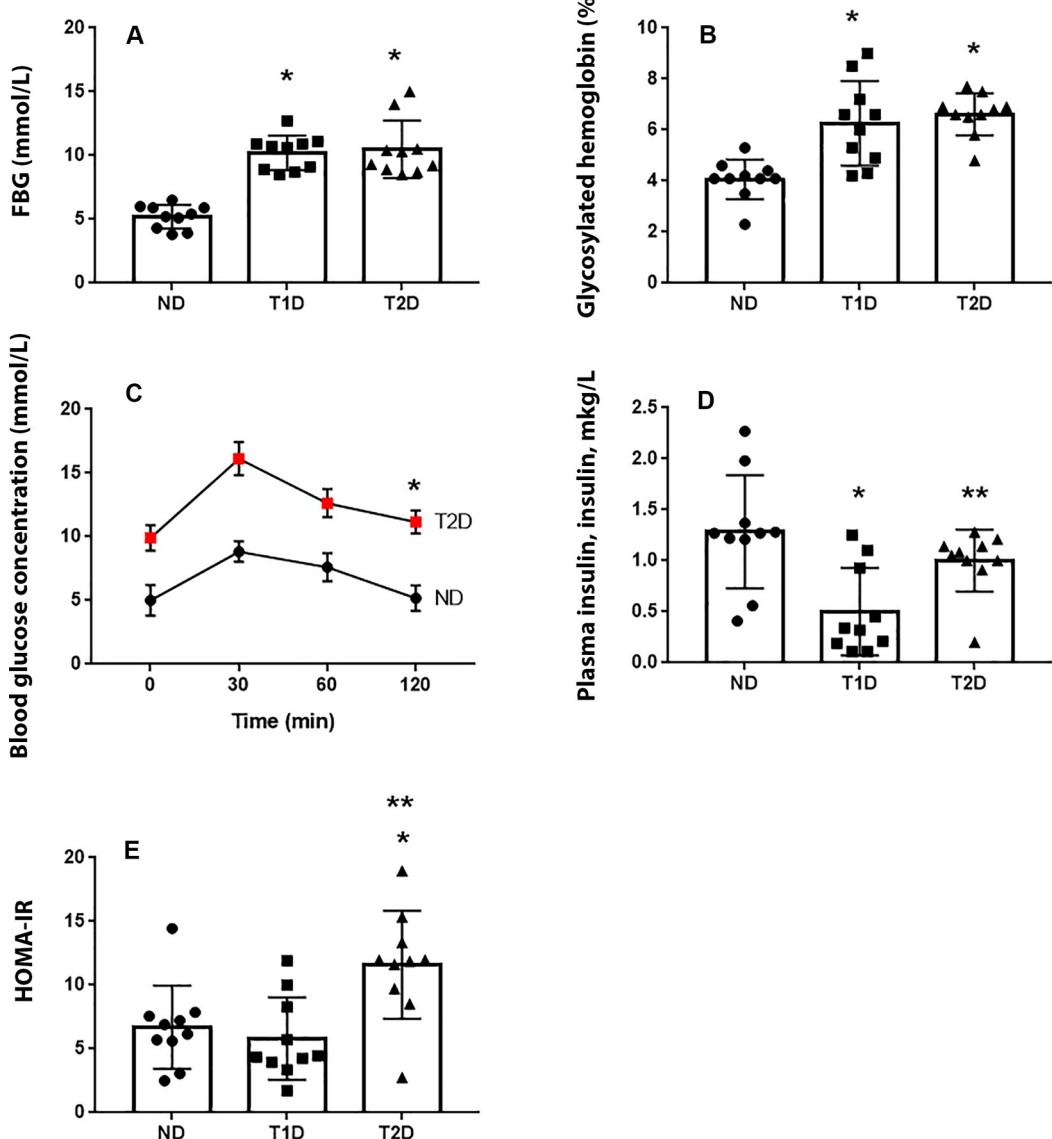

**Fig 1. Validation of T1D and T2D models in rats.** The fasting blood glucose level (A) and concentration of glycated hemoglobin (B) are similar in the both diabetic groups. T2D is characterized by poor glucose tolerance (), normal blood insulin concentration (D) and increased mean of HOMA-IR index (E). ND–non-diabetic group, T1D –type 1 diabetes, T2D –type 2 diabetes. Each data point represents one rat in each group. The error bars represent the standard error of the mean and the column represents the mean. *–p < 0.05 in comparison with ND group; **–p < 0.05 in comparison with T1D group.

Rats with T2D showed poor glucose tolerance, as evident by higher AUC value observed in T2D group compared to ND group (p < 0.05) (Fig 1) [38]. Blood insulin reduce only in T1D, in T2D this parameter was at the normal level (Fig 1D). Index HOMA-IR, used in clinical practice and scientific investigation to assess function of β-cells and insulin resistance and as a way of early T2D diagnosis [44], was calculated. As expected, HOMA-IR index was also increased in animals with experimental T2D compared to T1D group and non-diabetic animals (Fig 1E).

## Hepatic insulin-positive cells in the models of T1D and T2D

Insulin-positive cells (IPCs) were found in the liver tissue of both ND and diabetic rats (Fig 2), they were localized in the hepatic plates and along hepatic sinusoid spaces. The IPCs in hepatic plates correspond in structure, size and location to hepatocytes. In T1D IPCs were observed mainly in the peripheral area of hepatic plate, in T2D - in all areas. The number of insulin + hepatocytes was significantly higher in diabetic rats compared to ND counterparts. The largest number of these cells was observed in the rats with T2D, 10-fold higher compared to values found in ND rats (Table 2). Number of insulin-positive sinusoidal cells is also increased in the both diabetic groups, the largest number of insulin-positive sinusoidal cells were observed in T2D group.

## Markers of liver injury and inflammation in blood

To evaluate functional state of hepatocytes in the models of T1D and T2D, we analyzed activity of aminotransferases (ALT and AST), alkaline phosphatase (AP) and protein content in blood plasma (Fig 3A–3E). ALT is a rather specific marker of hepatocellular injury [45]. Increase in ALT level compared to normal was higher in T2D group than in T1D. A high ALT level led to a decrease in AST/ALT ratio (De Ritis coefficient) in T2D. There was an increase in serum AST level in T1D, compared to ND. In contrast, AST level in T2D did not change. No significant difference was observed in ALP level between ND and diabetic groups. Lower total protein was observed in T1D as compared to ND and T2D.

Taking into account inflammatory component in pathogenesis of diabetes, particularly in T2D, the leukocyte blood count was calculated. Relative to ND rats, the total number of white blood cells and lymphocytes in the peripheral blood of the T1D and T2D rats was elevated, especially in T2D. Granulocytes elevated only in T1D, demonstrating a high-scatter meaning in T2D (Fig 4A–4C).

## Histological and immunohistochemical evaluation of the liver

Diffuse-focal glycogen accumulation without signs of the toxic damage and rearrangement of liver histological architecture and vascular network was noted in both T1D and T2D (Fig 5).

Regeneration rates may reflect organ damage. Morphometric analysis showed an increase in the number of signs of both cellular and intracellular regeneration in the liver of diabetic animals, but differently in T1D and T2D. Thus, compared to ND animals, number of binucleated hepatocytes elevated in animals with T1D and T2D by 1.7 and 1.4-fold, and mitotic index by 1.7 and 2.0-fold respectively (Fig 6A and 6B).

It is known that sinusoids cells, first of all, macrophages and leukocytes, infiltrate damaged liver, are able to produce proinflammatory cytokines and growth factors and participate in regulation of destructive and compensatory processes as well as regenerative growth. Presumably, they can influence the formation of IPCs. As shown in Fig 7A and 7B, an increase in total number of sinusoidal cells and macrophages was observed in animals with both T1D and T2D compared to ND (p < 0.05).

Development of experimental T1D and T2D was accompanied by the rise in the number of CD45+ cells, located both in parenchyma and in perivascular spaces. It was noticed that in both T1D and T2D, most of the CD45+ cells are localized in liver parenchyma, while in ND animals their number in parenchyma and perivascular spaces was approximately the same (Fig 8A–8D). The number of the CD3+ cells was also elevated in the liver of diabetic animals. The most significant increase in the number of CD3+ cells is observed in liver parenchyma of animals with T2D, it exceeds control value by 15 times (Fig 8E–8H).

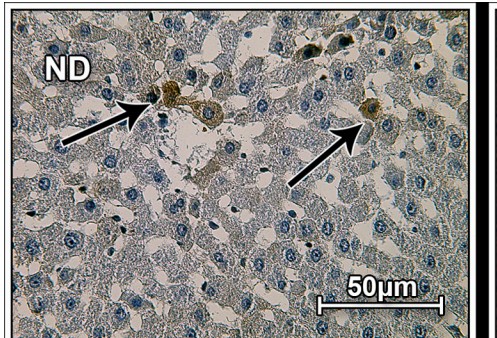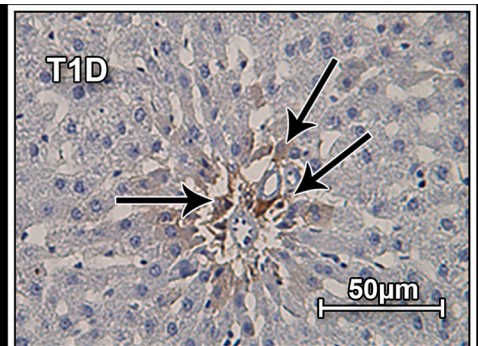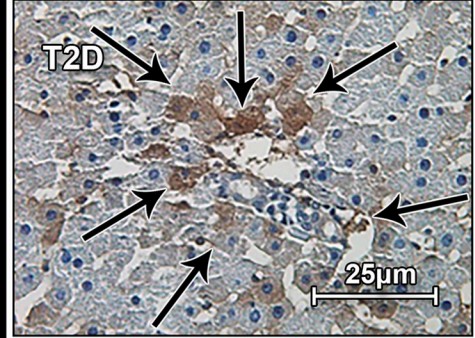

**Fig 2. Immunohistochemical (IHC) detection of proinsulin and insulin in liver tissue in rats.** Chromogenic IHC analysis showing brown staining in the cytoplasm, nuclei were counterstained with hematoxylin. Arrows show IPCs. Results demonstrate significant increase in the number of IPCs in the liver of T2D rats compared to ND and T1D liver samples. ND–non-diabetic group, T1D –type 1 diabetes, T2D –type 2 diabetes.

### Transcription factors Pdx1, MafA and Ngn3 in liver in T1D and T2D

Growth of the number of IPCs in the liver can be affected by transcription factors Pdx, MafA and Ngn3, which participate in reprogramming of cells in an adult body into IPCs and contributes to expression of insulin gene. In present study, Pdx1+, MafA+ and Ngn3+ cells were detected in liver of both ND and diabetic animals (Figs 9A, 10A and 11A). Compared to ND, the expression of MafA in liver heightened in both T1D and T2D, and Pdx1 –only in T1D ($p < 0.05$) (Figs 9B and 10B). Neither T1D nor T2D did not change the quantity of Ngn3 + hepatic cells (Fig 11B).

### Hepatic stem cells in the models of T1D and T2D

Reparative regeneration of damaged organs in adult body involves stem/progenitor cell, so we aimed to assess hepatic stem cell compartment. Quantitative analysis of the oval cell markers KRT19 and OV6 revealed, that number of KRT19+ and OV6+ cells increased in the liver of diabetic animals. Data presented in Fig 12A–12D showed that the greatest number of KRT19 + and OV6+ cells in liver were detected in animals with T2D.

## Discussion

Insulin+ cells found in various organs are attracting more and more attention of researchers, since they can partially compensate for damage of islet beta-cells in DM [6, 7]. Presence of insulin+ cells in liver have been established both in healthy and diabetic animals [6, 16, 17, 19,

**Table 2. Quantity of hepatic IPCs (mean ± SEM).**

| | ND | T1D | T2D |
|---|---|---|---|
| Insulin+ hepatocytes, N/mm² | 14.21 ± 1.69 | 26.56 ± 2.77 [1] | 144.17 ± 10.36 [1;2] |
| | | Among them: | |
| In peripheral area, N/mm² | 0.34 ± 0.09 | 15.56 ± 3.09 [1] | 40.12 ± 4.98 [1;2] |
| Insulin+ sinusoidal cells, N/mm² | 13.81 ± 0.85 | 20.82 ± 1.21 [1] | 39.04 ± 5.44 [1;2] |
| | | Among them: | |
| In peripheral area, N/mm² | 0.63 ± 0.03 | 5.28 ± 0.32 [1] | 19.03 ± 1.71 [1;2] |

[1]—$p < 0.05$ in comparison with ND

[2]—$p < 0.05$ in comparison with T1D. Data are mean ± SEM.

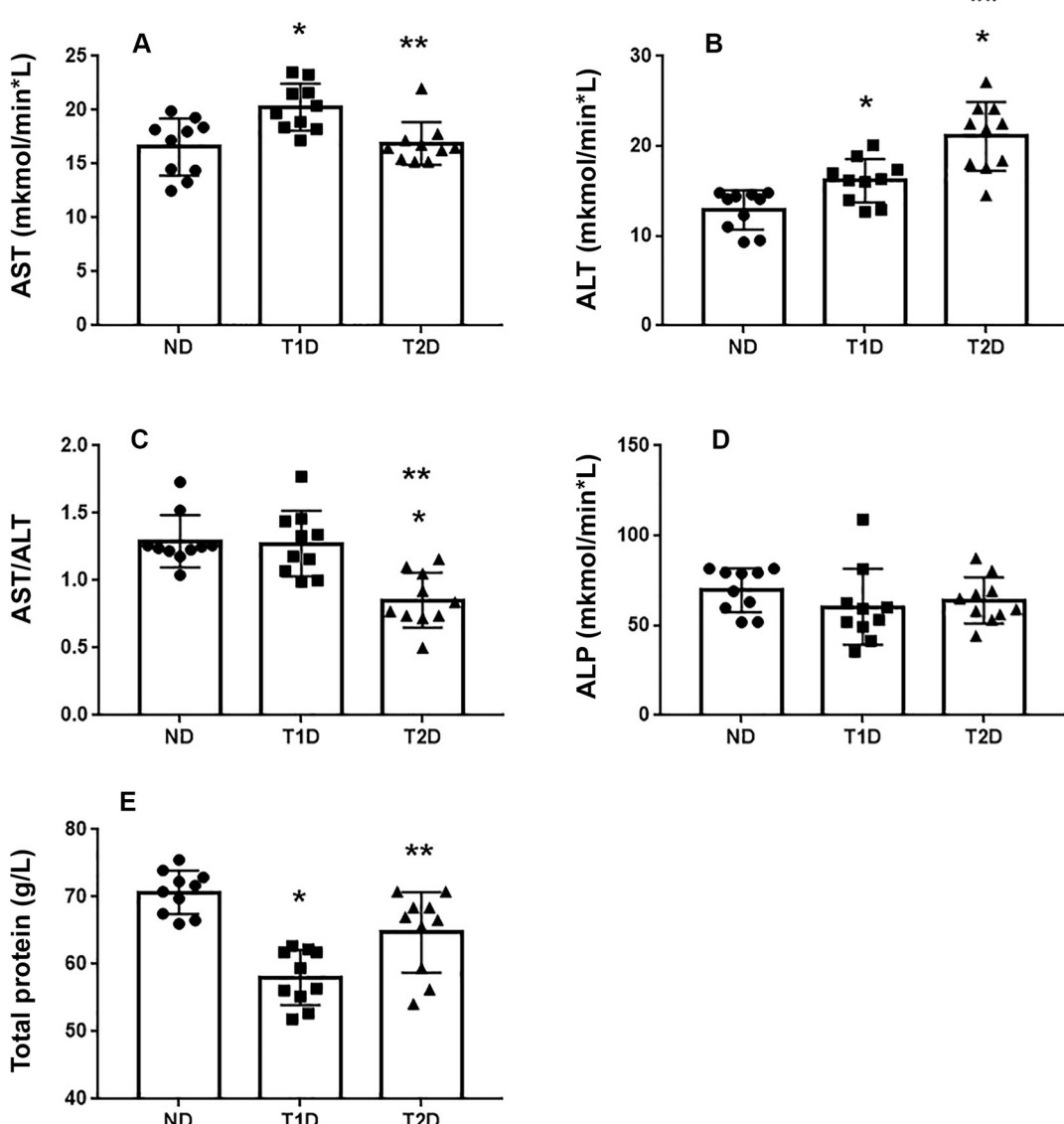

**Fig 3.** Quantitative examination of liver damage markers AST (A), ALT (B), AST/ALT (C), ALP (D) and total protein (E) in rats. Each data point represents one rat in each group. The error bars represent the standard error of the mean and the column represents the mean. ND–non-diabetic group, T1D –type 1 diabetes, T2D –type 2 diabetes.*–p < 0.05 in comparison with ND group; **–p < 0.05 in comparison with T1D group.

46]. Liver cells represent a potential target for conversion into insulin-producing cells for numerous reasons, including close origin from adjacent regions of the endoderm and relatedness of early embryonic stages with pancreas with expression of common regulatory expression factors [47].

In present study we evaluate the mass of hepatic IPCs in rat models of T1D and T2D and discuss the role of glycaemia level, expression of pancreatic transcription factors Pdx1, MafA and Ngn3, number of oval cells, inflammation and tissue damage in emerging of hepatic IPCs in vivo. We demonstrated that the mass of hepatic IPCs may be differ at equal glycemia level. The main findings are: 1) Hepatic insulin synthesis increased in rat models of T1D and T2D with the similar level of glycemia differently, the number of insulin+ cells in the liver in T2D is

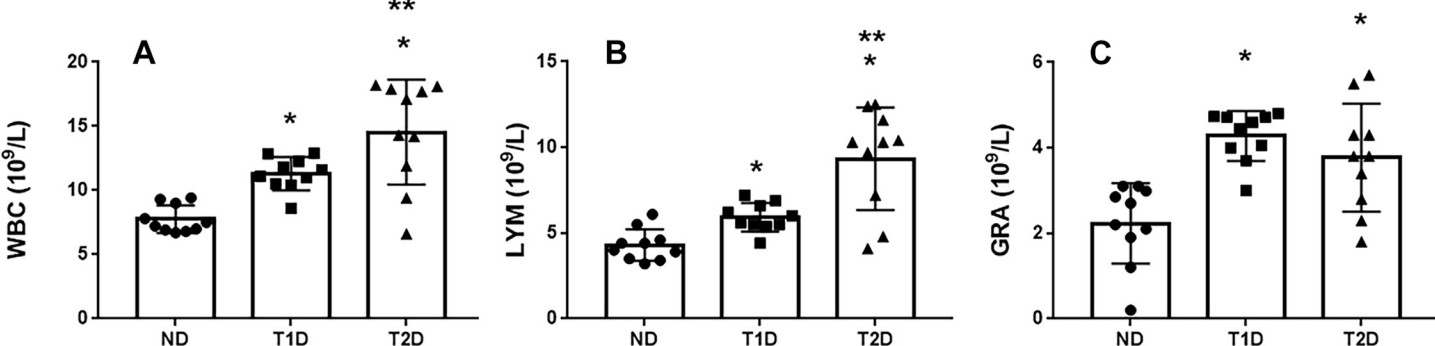

**Fig 4. Hematological profile of rats with experimental T1D and T2D.** The increasing of the inflammatory biomarkers amount - leukocytes (WBC) (A), lymphocytes (LYM) (B) and granulocytes (GRA) (C) - is detected in both T1D and T2D, however, in T2D the more prominent increasing of WBC and LYM is observed; cells in μl. ND–non-diabetic group, T1D –type 1 diabetes, T2D –type 2 diabetes. Each data point represents one rat in each group. The error bars represent the standard error of the mean and the column represents the mean. *–$p < 0.05$ in comparison with ND group; **–$p < 0.05$ in comparison with T1D group.

in several times more than in T1D. 2) Probably, generation of insulin+ cells in liver is associated with inflammatory agents.

## Blood glucose and hepatic insulin production in the models of T1D and T2D

The feasibility of controlling hyperglycemia in diabetes through hepatic synthesis of insulin is attractive for researchers. Glucose stimulates insulin gene transcription [48–50]. It was demonstrated that hyperglycemia either caused by diabetes or not associated with it led to the appearance of proinsulin+ and insulin+ cells in non-pancreatic tissues [6, 8]. In the present study, it was observed that number of insulin+ cells in liver is higher in animals with diabetes compared to ND control, consistently with [6, 8]. However, despite similar increased glucose and HbA1 levels in the blood of animals with experimental T1D and T2D, the number of insulin+ hepatocytes in animals with T1D and T2D was different. According to our observation, mass of hepatic IPCs were in several times more in T2D vs. T1D. Additionally, it was found that localization of insulin+ cells in the liver is different in T1D and T2D. Summarize, we suggest that hyperglycemia is not the single necessary for generation of insulin+ cells in liver. It consistent with previous foundations that normal liver cells did not produce insulin when stimulated by glucose alone. There is no definitive answer to the question of what stimuli other than

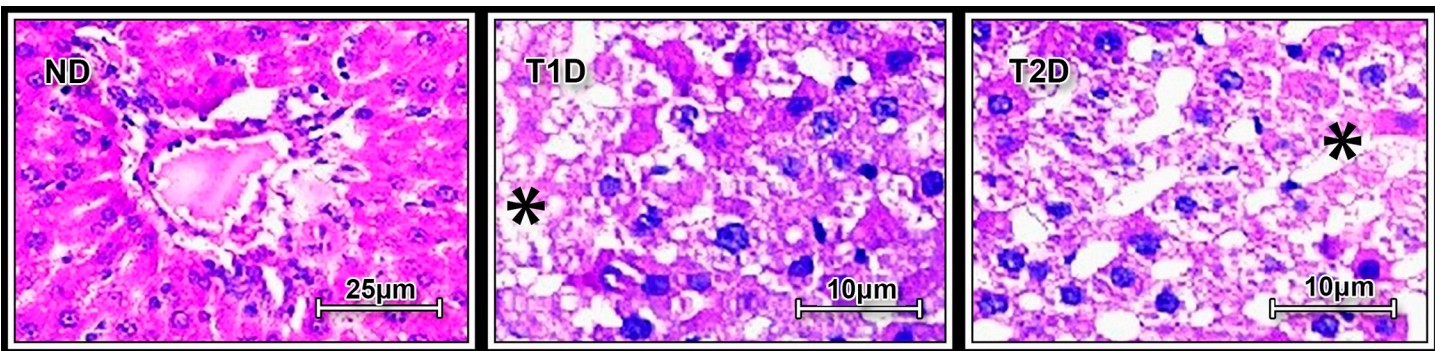

**Fig 5. Periodic acid-Schiff (PAS) staining of liver tissue in rats.** ND–non-diabetic group, T1D –type 1 diabetes, T2D –type 2 diabetes. Asterisks show glycogen accumulation in hepatocytes in T1D and T2D rats.

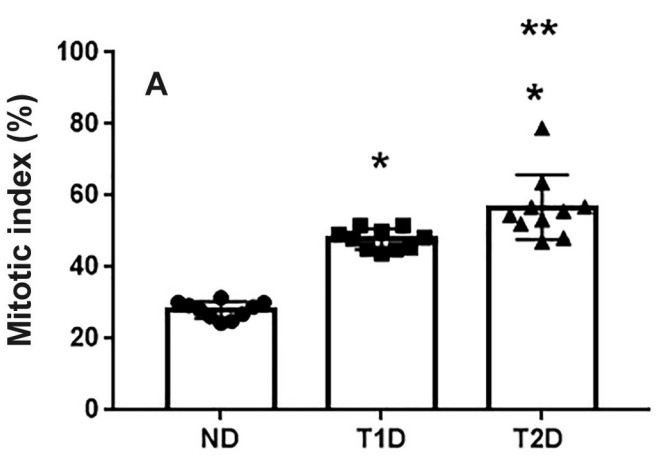
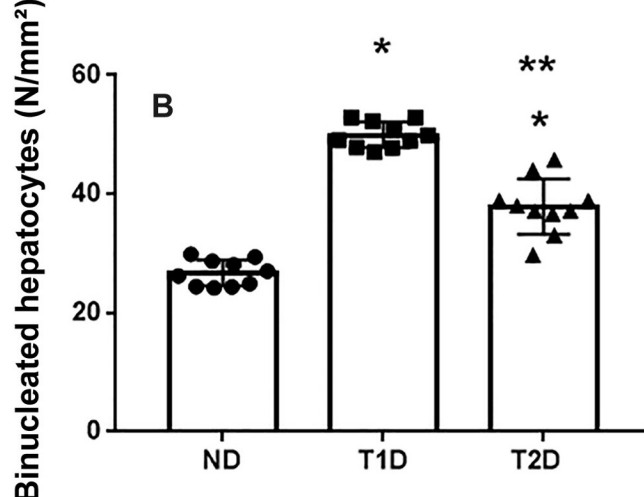

**Fig 6. Indicators of hepatic regeneration in rats.** (A): Mitotic index. (B): Number of binucleated hepatocytes per square millimeter of liver tissue. ND–non-diabetic group, T1D –type 1 diabetes, T2D –type 2 diabetes. Each data point represents one rat in each group. The error bars represent the standard error of the mean and the column represents the mean. *–p < 0.05 in comparison with ND group; **–p < 0.05 in comparison with T1D group.

hyperglycemia are necessary to induce insulin synthesis in the liver. In [17] was shown, that for synthesis of insulin in liver the simultaneous presence of both nitric oxide (NO), which is known as a signaling molecule that plays a key role in the pathogenesis of inflammation, and sugar were needed. Production of NO is necessary for stimulation of hepatic insulin secretion, triggered via adenoviral gene transfer of Pdx1, Neurod1 and MafA [51]. It was reported that glucose induced NO synthesis in the liver cells is capable of inducing Glut-4 synthesis and translocation in the hepatic cells and membrane for the expression of genetic elements leading to the synthesis of insulin [52]. So, hyperglycemia is a necessary, bur insufficient agent, stimulated hepatic insulin synthesis.

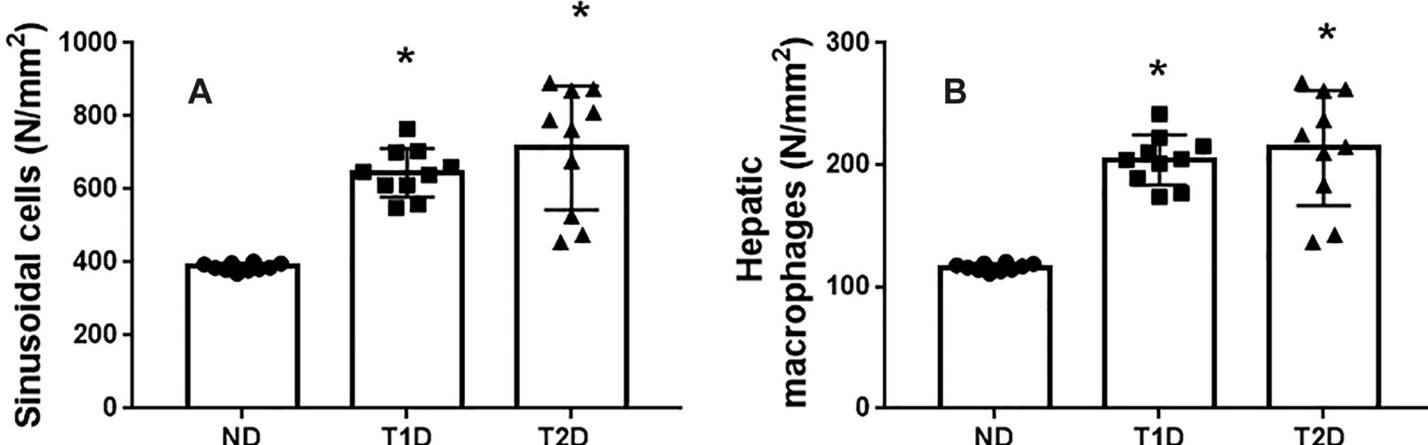

**Fig 7. Quantitative assessment of cells, maintaining immune homeostasis in liver tissue in rats.** The number of sinusoidal cells (A) and macrophages (B) are increased in both T1D and T2D. ND–non-diabetic group, T1D –type 1 diabetes, T2D –type 2 diabetes. Each data point represents one rat in each group. The error bars represent the standard error of the mean and the column represents the mean. *–p < 0.05 in comparison with ND group; **–p < 0.05 in comparison with T1D group.

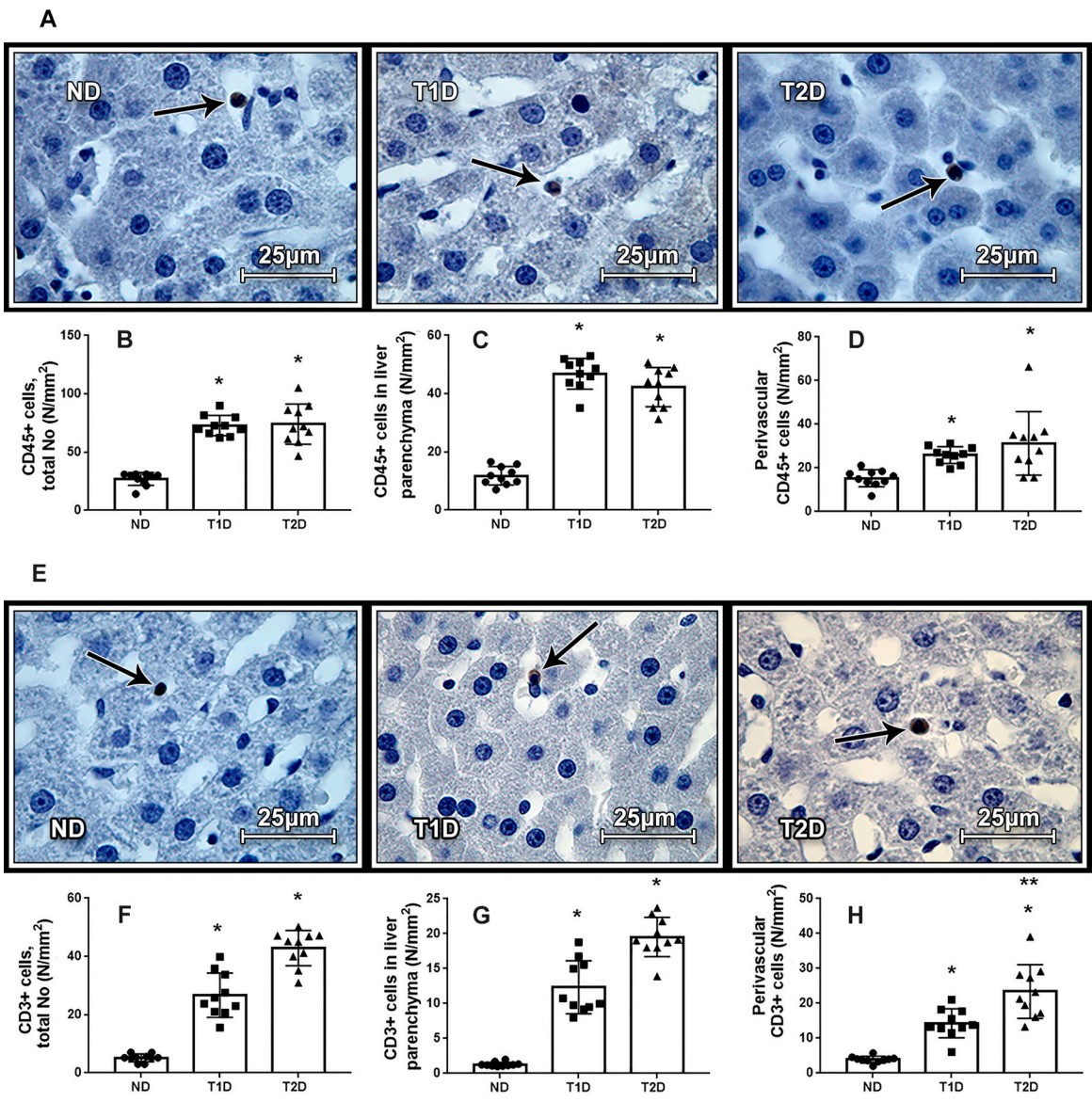

**Fig 8. Leukocytes in the liver in rats.** Chromogenic IHC analysis showing brown staining in the cytoplasm and membrane of CD45+ (A) and CD3+ (E) cells. Arrows show CD45+ (A) and CD3+ (E) cells, counterstaining with hematoxylin. Infiltration of liver tissue with CD45+ (B-D) and CD3+ (F-H) cells increased in both diabetic group. ND–non-diabetic group, T1D –type 1 diabetes, T2D –type 2 diabetes. Each data point represents one rat in each group. The error bars represent the standard error of the mean and the column represents the mean. *–p < 0.05 in comparison with ND group; **–p < 0.05 in comparison with T1D group.

## Hepatic injury and hepatic insulin function in the models of T1D and T2D

It is known, that organ injury trigger regeneration process to maintain tissue integrity and function [53]. We measured ALT, AST, ALP and total protein in blood to test liver function and hepatic protein synthesis. Increased ALT level and low AST/ALT ratio is associated with the insulin resistance [54, 55], which verifies T2D state. Total protein slightly decreased in the blood serum in T1D and T2D vs ND, which corresponds to the diabetic state [56, 57]. Activity of liver enzymes indicate that hepatocytes retain their functions and a relatively low level of damage. We suggest that observed increased number of hepatic IPCs in T1D and especially in T2D cannot be result of just liver injury.

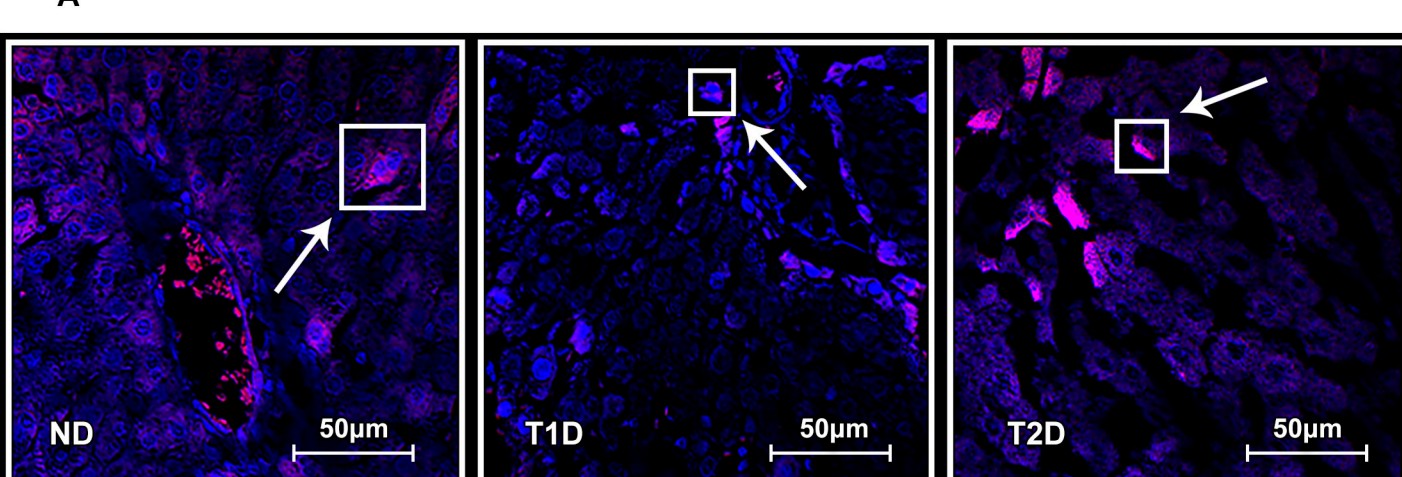

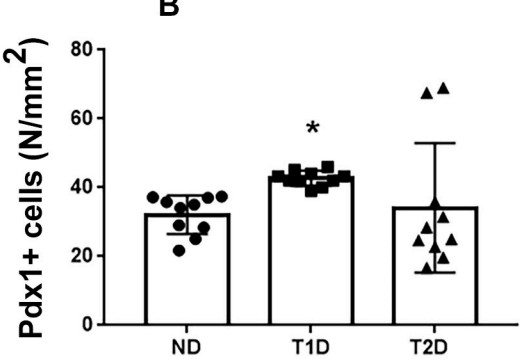

**Fig 9. Pdx1+ cells in liver tissue in rats.** (A): fluorescent microscopy of Pdx1 (magenta), (B): number of Pdx1+ hepatic cells. Arrows mark Pdx1+ cells. The nuclei stained with DAPI (blue). ND–non-diabetic group, T1D –type 1 diabetes, T2D –type 2 diabetes. Each data point represents one rat in each group. The error bars represent the standard error of the mean and the column represents the mean. *–p < 0.05 in comparison with ND group; **–p < 0.05 in comparison with T1D group.

Since the liver and pancreas have a common origin, we hypothesized that the generation of IPCs in liver can be stimulated by the same factors as the formation of IPCs in pancreas. Restoring of pancreatic beta-cells mass can occur through stimulation of existing beta-cell replication, activated by transcription factors transdifferentiation of specialized mature cells into beta-cells and from stem/progenitor cells [58, 59].

### Hepatic progenitor cells and hepatic insulin production in the models of T1D and T2D

Regenerative processes in liver may proceed in different ways, depending on the severity and type of damage, including via progenitor cells [60]. In healthy liver progenitor oval cells are present in limited quantity, and liver damage is accompanied by an increase in their number [31, 33]. Increased number of Ov6+ and KRT19+ cells, observed in diabetic animals, especially in T2D, corresponds [31, 33]. At the same time, increased mitotic index indicates that hepatocyte-mediated regeneration process proceed in liver of both T1D and T2D rats. It is known, that intra-hepatic precursor cells (oval cells) proliferate and generate lineage only in situations in which hepatocyte proliferation is blocked or delayed. We suggest, that relatively low hepatic injury, preserve of hepatic function and increasing of cell regeneration testify about minor

A

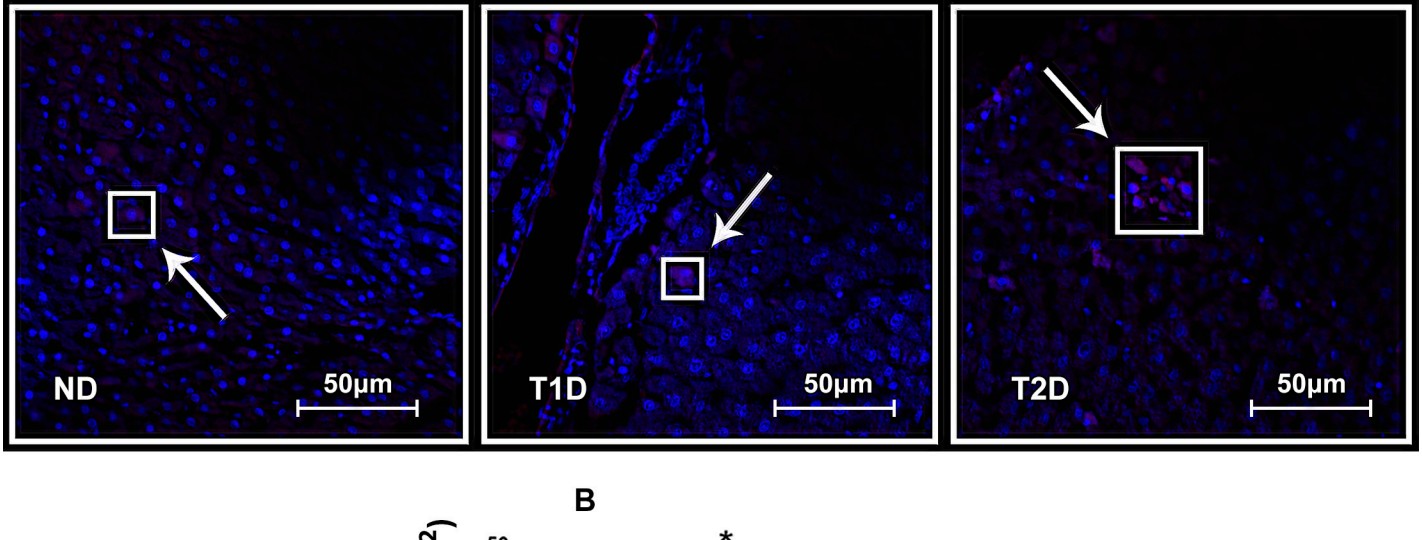

B

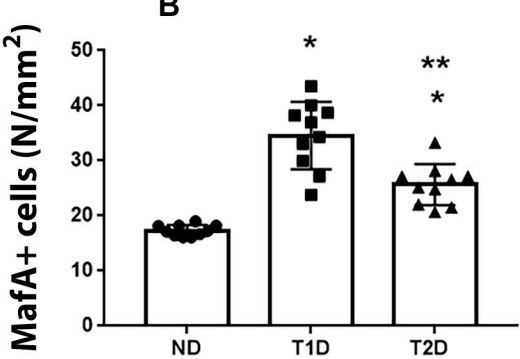

**Fig 10. MafA+ cells in liver tissue in rats.** (A): fluorescent microscopy of MafA (magenta), (B): number of MafA+ hepatic cells. Arrows mark Pdx1+ cells. The nuclei stained with DAPI (blue). ND–non-diabetic group, T1D –type 1 diabetes, T2D –type 2 diabetes. Each data point represents one rat in each group. The error bars represent the standard error of the mean and the column represents the mean. *–p < 0.05 in comparison with ND group; **–p < 0.05 in comparison with T1D group.

involving of stem/progenitor cells reserve in the generation of hepatic IPCs in the rats with experimental T1D and T2D.

## Transcription factors Pdx1, MafA, Ngn3 and hepatic insulin production in the models of T1D and T2D

Expression of the transcription factors Pdx1, Ngn3 and MafA (PNM) is able to reprogram not-beta cells to beta-like insulin-producing cells [25, 61–63]. We assessed quantity of Pdx1 + Ngn3+ and MafA+ cells in hepatic tissue of experimental rats. T1D and T2D did not significantly affect the expression of Ngn3, which corresponds to data obtained Al-Adsani et al. [63]. Ngn3 is reported as a pro-endocrine factor expressed in endocrine progenitor cells [64]. As reported, Ngn3 is sufficient for in vivo emergence of islet-like cells from hepatic progenitor cells, most likely oval cells, but not for transdifferentiations of hepatocytes [65]. However, earlier discussed, that low intensity of hepatic injury does not imply the involving of oval cells in the process of regeneration and neogenesis. We hypothesized that low level of Ngn3 expression correlates with the assumption of low involvement of the liver stem reserve in the formation of hepatic IPCs.

**A**

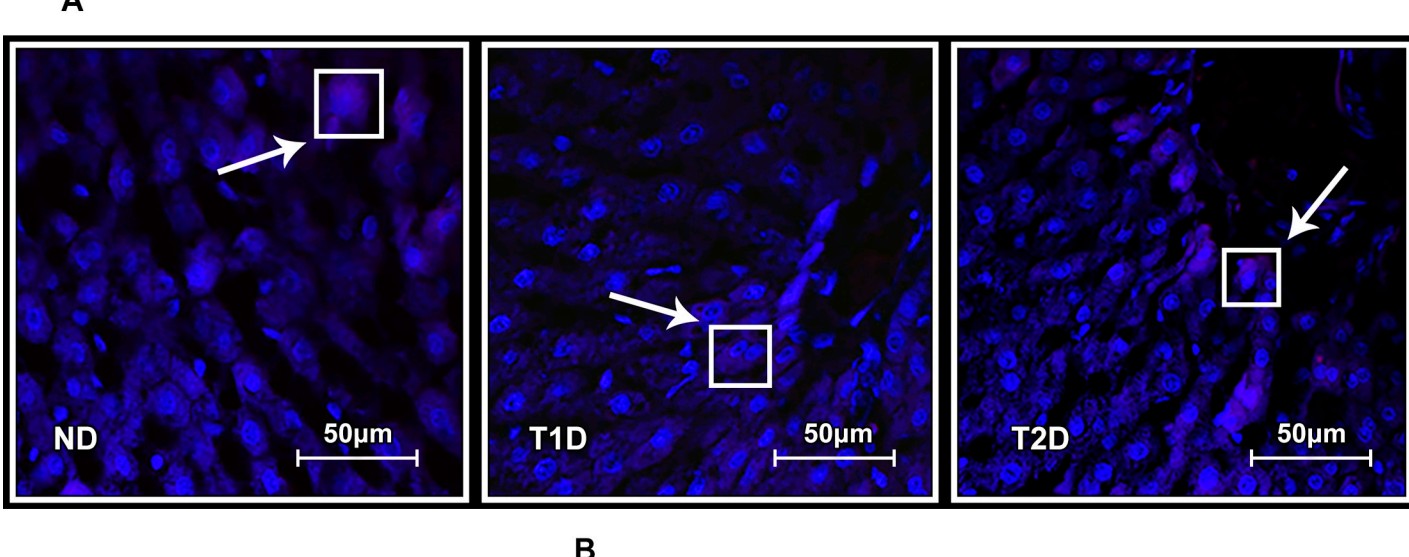

**B**

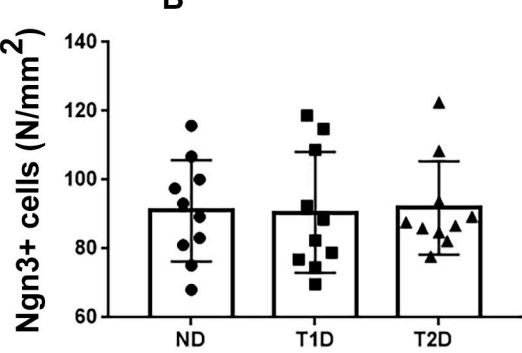

**Fig 11. Ngn3+ cells in liver tissue in rats.** (A): fluorescent microscopy of Ngn3 (magenta), (B): number of Ngn3+ hepatic cells. Arrows mark Pdx1+ cells. The nuclei stained with DAPI (blue). ND–non-diabetic group, T1D –type 1 diabetes, T2D –type 2 diabetes. Each data point represents one rat in each group. The error bars represent the standard error of the mean and the column represents the mean. *–$p < 0.05$ in comparison with ND group; **–$p < 0.05$ in comparison with T1D group.

Pdx1 regulates expression of various genes during early development of pancreas, including formation of beta-cells. Pdx1 is required for proliferation and maturation of beta-cells and is also important for regulation of insulin gene expression in them. MafA is required for functional activity of beta-cells, regulating synthesis of insulin and GLUT2. The importance of MafA for the activation of insulin genes and it expression in pancreatic beta-cells increases the likelihood that MafA is a major activator of pancreatic beta-cell formation and function [66]. In this regard, it can be assumed that an increase in the number of MafA+ cells indicates further transdifferentiation of these cells into insulin-producing ones.

T2D showed lower expression of Pdx1 and MafA in comparison with T1D. Less expression of Pdx1 and MafA in liver in T2D is associated with higher serum insulin vs T1D. Al-Adsani et al. reported, that expression of PNM transcription factors in the liver of recovering streptozotocin-induced diabetic rats decreased as blood insulin levels increased [63].

In [67] it was demonstrated that low levels of Pdx1 and MafA are needed for islet cell function and insulin release where maintained high levels of both genes lead to dysregulation in the metabolic processes.

Increasing in the number of IPCs occurs in the terms of different expression of PNM transcription factors which do not contradict with the findings of Al-Adsani et al. [63]. The greater

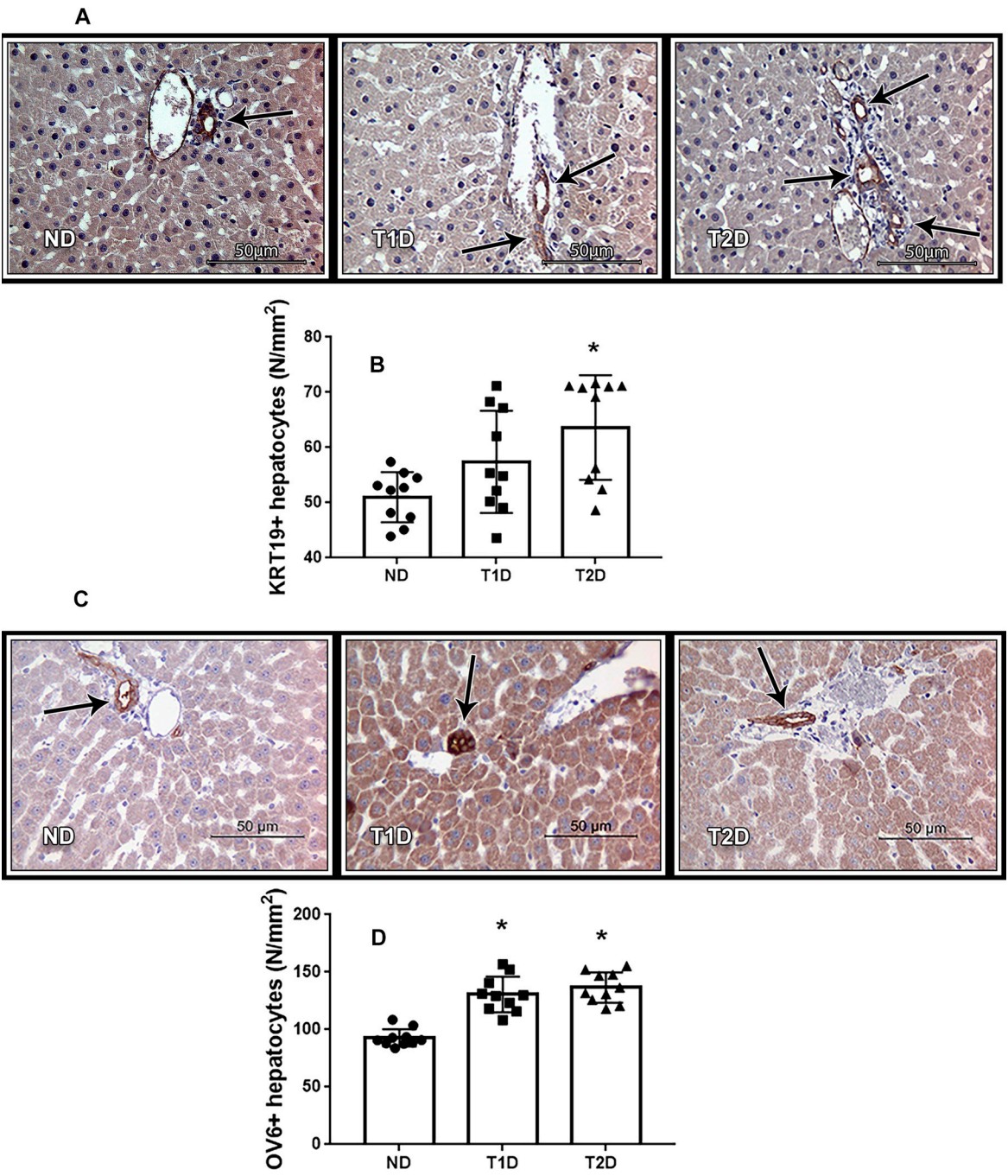

**Fig 12. Oval cells in liver tissue in rats.** Chromogenic IHC analysis showing brown staining in the cytoplasm of KRT19+ (A) and OV6+ (C) cells. Arrows show KRT19+ (A) and OV6+ (C) cells, counterstaining with hematoxylin. Quantitative analysis showed number of KRT19+ (B) and OV6+ (D) cells. ND–non-diabetic group, T1D –type 1 diabetes, T2D –type 2 diabetes. Each data point represents one rat in each group. The error bars represent the standard error of the mean and the column represents the mean. *–p < 0.05 in comparison with ND group; **–p < 0.05 in comparison with T1D group.

mass of hepatic IPCs at a lower number of Pdx+ cells in T2D compared with T1D suggests the presence of a mediator between the expression of Pdx1 and the expression of the insulin gene. It is likely that this mediator can be activated by inflammation. This is consistent with the previous investigations [17, 35, 52].

## Inflammation and hepatic insulin production in the models of T1D and T2D

Chronic inflammation is a hallmark of both T1D and T2D [68, 69]. Significant increase in such hematological parameters as the number of leukocytes, lymphocytes and granulocytes combined with the increased number of CD45+ cells in the liver in T1D and T2D compared to healthy control indicate the inflammatory process as well at the level of the whole body as in the liver. Increasing content of the CD45+ cells in perivascular areas and liver parenchyma corresponds to exudative phase of inflammation. However, in the model of T2D we observe higher severity of inflammatory process and insulin resistance (detected by increasing HOMA-IR index), which is also associated with inflammation. Increased mass of hepatic IPCs is accompanied with pronounced lymphocytosis and increased hepatic infiltration with CD3 + cells in the T2D. Taking into attention morphogenetic activity of lymphocytes, it may assume involving of lymphocytes in the generation of hepatic IPCs.

Pro-inflammatory stress is associate with epigenetic changes [70] and can activate multiple gene expression [71, 72]. Recent evidence suggests that inflammation may induced chromatin changes, which may play an important role in gene reprogramming [73]. It was showed that chronic inflammatory conditions may shape cell fate decision [74]. Also was demonstrated, that metabolic shifts can occur in cells as a response to infection and inflammation [75].

So, it was suggested, that proinflammatory microenvironment via multiple contact- and paracrine- mediated interactions shape the cell differentiations in different tissues–pancreatic cells [76], adipose tissue [74], immune cells [77], stromal cells [78].

Earlier we supposed, that inflammation in the pancreas caused by diabetic agent streptozotocin first triggers processes aimed at restoring the functional integrity of the pancreatic tissue, and, combined with hyperglycemia, affect the generation of insulin+ cells in exocrine pancreas. It manifest itself in increased mass and square of extra-islet pancreatic cells in 30 days of experimental T2D in rats. We demonstrated, that progress of inflammation in diabetes develops is associated with the dysfunction of extra-islet insulin+ cells, but decreasing the degree of macrophage infiltration and inflammatory level in pancreatic exocrine tissue in experimental model of T2D lead to the increasing of the mass of extra-islet insulin+ cells [35].

In present study we note that increased number of hepatic IPCs in T2D vs T1D is associated with higher severity of inflammation in T2D. At the same time, all other factors traditionally considered as affecting the ectopic expression of insulin gene (hyperglycemia, liver injury, expression of transcription factors PNM, involving of stem/progenitor cells) in T2D compare to T1D is similar or even less. We speculate, that hepatic insulin synthesis is stimulated at a certain inflammatory level, decrease in both insufficiency and excessiveness. In this case, increased hepatic insulin expression in experimental T1D and T2D may be the part of the manifestation of cellular stress, a typical cellular response to any form of macromolecular damage aimed at restoring cellular and tissue homeostasis [79, 80]. If we consider ectopic insulin expression from the position of cellular stress, it is natural to assume the occurrence of extra-pancreatic IPCs not only in diabetes, but also in a wide spectrum of different pathologies, which, in our opinion, is a promising direction of future investigations.

## Conclusion

We have demonstrated that mass of hepatic insulin+ cells in rat models of T1D and T2D with the similar glycemia is differ. We concluded, that hepatic expression of insulin activated by the combined action of a number of factors, among which the most important seems to be inflammation, and occur differently in T1D and T2D. Since extra-pancreatic insulin+ cells can partially compensate for lack of insulin in diabetes, it is very important to clear metabolic

regulation of insulin expression. The results suggested that development of anti-inflammatory-focused therapy for diabetes is reasonable and promising.

## Supporting information

**S1 File.**
(DOCX)

## Author Contributions

**Conceptualization:** Ksenia Sokolova, Irina Danilova.

**Data curation:** Ksenia Sokolova, Irina Danilova, Madina Baykenova, Irina Gette.

**Funding acquisition:** Musa Abidov.

**Investigation:** Ksenia Sokolova, Madina Baykenova, Irina Gette.

**Methodology:** Ksenia Sokolova, Irina Danilova, Madina Baykenova, Irina Gette, Elena Mychlynina.

**Project administration:** Burcin Aydin Ozgur.

**Resources:** Musa Abidov, Irina Danilova.

**Supervision:** M. Temel Yilmaz.

**Validation:** Irina Danilova.

**Writing – original draft:** Ksenia Sokolova, Irina Danilova, Madina Baykenova.

**Writing – review & editing:** Ksenia Sokolova, Irina Danilova, Burcin Aydin Ozgur, Ali Osman Gurol, M. Temel Yilmaz.

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
