## [Decision Letter · Decision Letter 0]

6 Aug 2023

PONE-D-23-22087Hepatic Insulin Synthesis Increases in Rat Models of Diabetes Mellitus Type 1 and 2 DifferentlyPLOS ONE

Dear Dr. Aydin Ozgur,

Thank you for submitting your manuscript to PLOS ONE. After careful consideration, we feel that it has merit but does not fully meet PLOS ONE’s publication criteria as it currently stands. Therefore, we invite you to submit a revised version of the manuscript that addresses the points raised during the review process. Please submit your revised manuscript by Sep 20 2023 11:59PM. If you will need more time than this to complete your revisions, please reply to this message or contact the journal office at plosone@plos.org. Please include the following items when submitting your revised manuscript:A rebuttal letter that responds to each point raised by the academic editor and reviewer(s). You should upload this letter as a separate file labeled 'Response to Reviewers'.A marked-up copy of your manuscript that highlights changes made to the original version. You should upload this as a separate file labeled 'Revised Manuscript with Track Changes'.An unmarked version of your revised paper without tracked changes. You should upload this as a separate file labeled 'Manuscript'.

We look forward to receiving your revised manuscript.

Kind regards,

Dr Riham M. Aly

Academic Editor

PLOS ONE

Journal Requirements:

   "122020900136-4 IIP UB RAS" 

5. Please expand the acronym “IIP UB RAS” (as indicated in your financial disclosure) so that it states the name of your funders in full.

7. Your ethics statement should only appear in the Methods section of your manuscript. If your ethics statement is written in any section besides the Methods, please move it to the Methods section and delete it from any other section. Please ensure that your ethics statement is included in your manuscript, as the ethics statement entered into the online submission form will not be published alongside your manuscript. 

8. Please include a copy of Table 2 which you refer to in your text on page 24.

9. Please upload a copy of Supporting Information Table 2 which you refer to in your text on page 14.

Reviewers' comments:

Reviewer's Responses to Questions

**Comments to the Author**

1. Is the manuscript technically sound, and do the data support the conclusions?

Reviewer #1: Partly

Reviewer #2: Yes

2. Has the statistical analysis been performed appropriately and rigorously? 

Reviewer #1: Yes

Reviewer #2: Yes

3. Have the authors made all data underlying the findings in their manuscript fully available?

Reviewer #1: No

Reviewer #2: Yes

4. Is the manuscript presented in an intelligible fashion and written in standard English?

Reviewer #1: No

Reviewer #2: No

5. Review Comments to the Author

Reviewer #1: The manuscript "Hepatic Insulin Synthesis Increases in Rat Models of Diabetes Mellitus Type 1 and 2 Differently” investigates insulin positive cells in the liver by different methods in both types of diabetes mellitus which is interesting; however, I think this manuscript and its figures need major revision.

General comments:

• Manuscript needs English language revising

• Paragraph should not start with abbreviation, so please write the full name, and here are some examples: IPS “third paragraph in introduction”, T1D “second paragraph in material and methods”, …ect.

Abstract:

• Line 6; Replace were not (done) with were not (performed).

Introduction:

• Correct the following words: per-manent to permanent, idi-opathic, proinsulin-, ex-pression, in-sulin, vec-tor, peculiar-ities, insu-lin, transdif-ferentiate, insu-lin, phago-, immunolog-ical, pit-cells, inflammation – in,

• Introduction is too long, there is no clear coherence and sequence in the writing, and most of the paragraphs require reconstruction and here are some examples for improvement:

Old one: In any case, the number of the functioning β-cells is insufficient in both T1D and T2D, and β-cells loss and dysfunction are the main reasons of hyperglycemia and associated with it complications [5].

New one: Loss and dysfunction of β-cells are the main reasons of hyperglycemia and its associated complications in both types of DM [5].

Old one: first, they are an additional source of insulin, the increase in the significance of which for the organism is logical to assume when pancreatic islets β-cells are damaged. Earlier we demonstrated [8] that in experimental T2D mass and functional activity of islet β-cells decrease faster, than of IPCs in pancreatic acini and ducts. Understanding the mechanisms and conditions, leading to the generation of IPCs in organs, are able to alleviate the problem of lack of β-cells. Second, since knowledge about ectopic insulin expression and, more, developmental plasticity of adult somatic cells and its reprogramming to committed lineages, unsufficient for complete and clear picture of this phenomenon, studying of extra-pancreatic insulin expression is also of theoretical interest.

New one: first, they are additional sources of insulin, as they showed significant increase within organism that are able to alleviate the problem when pancreatic islets β-cells are damaged as recorded in our previous study [8]. Second, to disclose the extra pancreatic insulin expression from adult somatic cells and reprogramming it to committed lineages.

• Write the full name of the Pdx1 abbreviation within the text.

• Rewrite the aim of the study in a simple and clear form.

Material and methods:

• Correct the following words: ac-cording, fluo-rescent.

• Insulin+, macrophages (F4/80+ cells), oval cells (OV6+ and KRT19+ cells), CD3+ and CD45+ cells were detected by IHC, (replace and by while) Pdx1+, MafA+, Ngn3+ cells were detected using fluorescent IHC.

• Add data about the color and site of immune-expression (cytoplasmic or nuclear) in the stained cells.

• Morphometric analysis: specify the number of captured field from each slide/group.

RESULTS

Result section needs SUBSTANTIAL revision.

• Correct the following words: ob-served, insu-lin, De-pending, ana-lyzed, pro-tein, particu-larly, rear-rangement, regenera-tive,

• Write the full name of the following abbreviations within the text (AUC, HOMA-IR).

• Figure (1 C) compares ND and T2D, while figure legend compares T2D from T1D, revise it.

• Results of (Figure 1 from F to J) are not mentioned in figure legend, need to be added.

• Results of (Figure 2 A, C to I) are not mentioned in the text, need to be added.

• Add scale bar in all photomicrographs in figure 2.

• Reorganize the writing of result section (text and legend) according to figures distribution. For example authors write about figure 1 A-D, then fig 2B, then figure 6, then return to figure 1F-J and so on…...

• For scientific benefit, it is better to insert each bar charts immediately below their corresponding histological or IHC figures.

• Figures 5 and 6 are the same figure???? Need to insert figure 5 again.

• Figure 6 formed of only five bar charts (A-E), while in result section authors had mention results about (fig 6 G&F)???

• Hepatic Stem Cells in the Models of T1D and T2D: (Fig. 6A-B) NOT (Fig. 6C-D) showed that the greatest number of OV6+ and KRT19+ cells in liver. Correct.

Discussion:

• Discussion section needs to be re-written in a better way and most of paragraphs need to be reconstructed.

• Correct the following words: an-imals, insu-lin, ex-pression, gen-eration, in-flammation, dia-betes, an-imals, gen-eration, insu-lin, in-crease, con-trol, re-storing, ex-tra).

• Remove the sentence: (In this study, for the first time, we demonstrated difference in the number of IPCs in the models of T1D and T2D.), since there is a previous study investigated the same points of the current research https://doi.org/10.1016/j.scr.2020.101958

• Remove this repeated sentence in the 2nd paragraph in discussion (Glucose is the major physiologic regulator of insulin gene expression.) as it was mentioned before in the 4th paragraph in introduction section.

• Write the full name of the following abbreviations within the text (Glut, PNM,

• Correct the following sentence: So, hyperglycemia is a necessary, but insufficient agent, in stimulation of hepatic insulin synthesis, instead of (bur unsufficient agent, stimulated hepatic insulin synthesis.).

Reviewer #2: The authors examined the quantity of IPCs in the liver could increase in rats both with T1D and T2D compared to non-diabetic controls. The ectopic hepatic insulin positive cells could be a new therapeutic window in T1D and T2D.

Major: E,F,G in Figure.2 was too hard to see or calculate the positive cells. The authors should replace the new or visuable pictures.

Minor:

There were some mistake letter such as "insu-lin-producing cells" and "immunolog-ical reactions". The authors should caefully revise these letter in the manuscript. Also, English grammer should be revised.

6. PLOS authors have the option to publish the peer review history of their article (what does this mean?). If published, this will include your full peer review and any attached files.

Reviewer #1: No

Reviewer #2: No

---

## [Author Response · Author response to Decision Letter 0]

27 Sep 2023

Dear colleagues, 

We would like to thank you for your helpful comments. Please finf below a point-by-point response to your comments and questions.

Response to Academic Editor Comments:

Point 1: 

Response 1: We ensure, that manuscript meets PLOS ONE's style requirements, including those for file naming.

Point 2:

Please note that funding information should not appear in any section or other areas of your manuscript. We will only publish funding information present in the Funding Statement section of the online submission form. Please remove any funding-related text from the manuscript.

Response 2:

We remove any funding-related text from the manuscript.

Point 3: 

We note that the grant information you provided in the ‘Funding Information’ and ‘Financial Disclosure’ sections do not match. 

Response 3:

Done. 

Point 4: 

Thank you for stating the following financial disclosure: "122020900136-4 IIP UB RAS" 

Response 4:

Done.

Point 5: 

Please expand the acronym “IIP UB RAS” (as indicated in your financial disclosure) so that it states the name of your funders in full.

Response 5:

Done.

Point 6: 

In your Data Availability statement, you have not specified where the minimal data set underlying the results described in your manuscript can be found. PLOS defines a study's minimal data set as the underlying data used to reach the conclusions drawn in the manuscript and any additional data required to replicate the reported study findings in their entirety. All PLOS journals require that the minimal data set be made fully available. For more information about our data policy, please see http://journals.plos.org/plosone/s/data-availability.

Response 6:

Done. Minimal underlying data set was upload as a separate Supporting Information file.

Point 7: 

Your ethics statement should only appear in the Methods section of your manuscript. If your ethics statement is written in any section besides the Methods, please move it to the Methods section and delete it from any other section. Please ensure that your ethics statement is included in your manuscript, as the ethics statement entered into the online submission form will not be published alongside your manuscript. 

Response 7:

Done.

Point 8: 

Please include a copy of Table 2 which you refer to in your text on page 24.

Response 8:

Done. We inserted Tables 1 and 2 immediately after the first paragraph in which they are cited.

Point 9: 

Please upload a copy of Supporting Information Table 2 which you refer to in your text on page 14.

Done. We inserted Tables 1 and 2 immediately after the first paragraph in which they are cited.

Response to Reviewer 1 Comments:

Point 1: 

General comments: Manuscript needs English language revising.

Response 1:

Done.

Point 2:

Paragraph should not start with abbreviation, so please write the full name, and here are some examples: IPS “third paragraph in introduction”, T1D “second paragraph in material and methods”, …ect.

Response 2:

Done.

Point 3:

Abstract: Line 6; Replace were not (done) with were not (performed).

Response 3:

Done. 

Point 4:

Introduction: Correct the following words: per-manent to permanent, idi-opathic, proinsulin-, ex-pression, in-sulin, vec-tor, peculiar-ities, insu-lin, transdif-ferentiate, insu-lin, phago-, immunolog-ical, pit-cells, inflammation – in

Response 4:

Done.

Point 5:

Introduction is too long, there is no clear coherence and sequence in the writing, and most of the paragraphs require reconstruction and here are some examples for improvement:

Old one: In any case, the number of the functioning β-cells is insufficient in both T1D and T2D, and β-cells loss and dysfunction are the main reasons of hyperglycemia and associated with it complications [5].

New one: Loss and dysfunction of β-cells are the main reasons of hyperglycemia and its associated complications in both types of DM [5].

Old one: first, they are an additional source of insulin, the increase in the significance of which for the organism is logical to assume when pancreatic islets β-cells are damaged. Earlier we demonstrated [8] that in experimental T2D mass and functional activity of islet β-cells decrease faster, than of IPCs in pancreatic acini and ducts. Understanding the mechanisms and conditions, leading to the generation of IPCs in organs, are able to alleviate the problem of lack of β-cells. Second, since knowledge about ectopic insulin expression and, more, developmental plasticity of adult somatic cells and its reprogramming to committed lineages, unsufficient for complete and clear picture of this phenomenon, studying of extra-pancreatic insulin expression is also of theoretical interest.

New one: first, they are additional sources of insulin, as they showed significant increase within organism that are able to alleviate the problem when pancreatic islets β-cells are damaged as recorded in our previous study [8]. Second, to disclose the extra pancreatic insulin expression from adult somatic cells and reprogramming it to committed lineages.

Response 5:

We shortened and reconstructed the Introduction. 

Point 6:

Introduction: Write the full name of the Pdx1 abbreviation within the text.

Response 6:

Done.

Point 7:

Introduction: Rewrite the aim of the study in a simple and clear form.

Response 7:

Done. 

Point 8:

Material and methods: Correct the following words: ac-cording, fluo-rescent.

Response 8:

Done.

Point 9:

Material and methods: Insulin+, macrophages (F4/80+ cells), oval cells (OV6+ and KRT19+ cells), CD3+ and CD45+ cells were detected by IHC, (replace and by while) Pdx1+, MafA+, Ngn3+ cells were detected using fluorescent IHC.

Response 9:

Done.

Point 10:

Material and methods: Add data about the color and site of immune-expression (cytoplasmic or nuclear) in the stained cells.

Response 10:

Done. We specify the color and site of immune expression in the Figure Captions. 

Point 11:

Material and methods: Morphometric analysis: specify the number of captured field from each slide/group.

Response 11:

The number of captured field was specified in the block “Morphometric analysis”: All morphometric evaluations were based on 20 microscopy fields per tissue section of each rat and were converted to the 1 mm2 of liver parenchyma (N/mm2).

Point 12:

RESULTS: Result section needs SUBSTANTIAL revision. Correct the following words: ob-served, insu-lin, De-pending, ana-lyzed, pro-tein, particu-larly, rear-rangement, regenera-tive

Response 12:

Done.

Point 13:

RESULTS: Write the full name of the following abbreviations within the text (AUC, HOMA-IR).

Response 13: 

HOMA-IR – Done.

The full name of the abbreviation “AUC” there is in line # 313: The area under the curve (AUC) was calculated according to Sakaguchi et al. (2016) using the trapezoidal approximation of PG levels [38].

Point 14:

RESULTS: Figure (1 C) compares ND and T2D, while figure legend compares T2D from T1D, revise it.

Response 14:

Done.

Point 15:

RESULTS: Results of (Figure 1 from F to J) are not mentioned in figure legend, need to be added.

Response 15:

Done. 

Point 16:

RESULTS: Results of (Figure 2 A, C to I) are not mentioned in the text, need to be added.

Response 16:

Done. 

Point 17:

RESULTS: Add scale bar in all photomicrographs in figure 2.

Response 17:

Done.

Point 18:

RESULTS: Reorganize the writing of result section (text and legend) according to figures distribution. For example authors write about figure 1 A-D, then fig 2B, then figure 6, then return to figure 1F-J and so on…...

Response 18:

Done.

Point 19:

RESULTS: For scientific benefit, it is better to insert each bar charts immediately below their corresponding histological or IHC figures.

Response 19:

Done. 

Point 20:

RESULTS: Figures 5 and 6 are the same figure???? Need to insert figure 5 again.

Response 20:

Done.

Point 21:

RESULTS: Figure 6 formed of only five bar charts (A-E), while in result section authors had mention results about (fig 6 G&F)???

Response 21:

Done. All figures were remaked.

Point 22:

RESULTS: Hepatic Stem Cells in the Models of T1D and T2D: (Fig. 6A-B) NOT (Fig. 6C-D) showed that the greatest number of OV6+ and KRT19+ cells in liver. Correct.

Response 22:

We fixed in Figure Caption.

Point 23:

Discussion: Discussion section needs to be re-written in a better way and most of paragraphs need to be reconstructed.

Response 23:

Done.

Point 24:

Discussion: Correct the following words: an-imals, insu-lin, ex-pression, gen-eration, in-flammation, dia-betes, an-imals, gen-eration, insu-lin, in-crease, con-trol, re-storing, ex-tra).

Response 24:

Done.

Point 25:

Discussion: Remove the sentence: (In this study, for the first time, we demonstrated difference in the number of IPCs in the models of T1D and T2D.), since there is a previous study investigated the same points of the current research https://doi.org/10.1016/j.scr.2020.101958

Response 25:

The https://doi.org/10.1016/j.scr.2020.101958 research is the first study have demonstrated that functional insulin-producing cells (IPCs) can be generated by using induced pluripotent stem cells (iPSCs) derived from as well non-diabetic (ND) as T1D and T2D individuals in vitro. 

In our research we evaluate the mass of IPCs in the liver in the rat models of T1D and T2D with the comparable glycemic level in vivo and try to evaluate factors impact emerging of IPCS in the liver. 

Point 26:

Discussion: Remove this repeated sentence in the 2nd paragraph in discussion (Glucose is the major physiologic regulator of insulin gene expression.) as it was mentioned before in the 4th paragraph in introduction section.

Response 26:

Done.

Point 27:

Discussion: Write the full name of the following abbreviations within the text (Glut, PNM).

Response 27:

Done.

Point 28:

Discussion: Correct the following sentence: So, hyperglycemia is a necessary, but insufficient agent, in stimulation of hepatic insulin synthesis, instead of (bur unsufficient agent, stimulated hepatic insulin synthesis.).

Response 28:

Done.

Response to Reviewer 2 Comments:

Point 1 (Major): E,F,G in Figure.2 was too hard to see or calculate the positive cells. The authors should replace the new or visuable pictures.

Response 1:

Done.

Point 2 (Minor):

There were some mistake letter such as "insu-lin-producing cells" and "immunolog-ical reactions". The authors should caefully revise these letter in the manuscript. Also, English grammer should be revised.

Response 2:

Done.

List of changes 

1) We revised the numbering of affiliations. The right variant is:

Musa Abidov 1, Ksenia Sokolova 2, Irina Danilova 2, Madina Baykenova 3, Irina Gette 2, Elena Mychlynina 2, Burcin Aydin Ozgur 4,5* , Ali Osman Gurol 6,7, M. Temel Yilmaz 8 

1 Institute of Immunopathology and Preventive Medicine, Ljubljana, Slovenia

2 Institute of Immunology and Physiology, Ural Branch of the Russian Academy of Sciences, Yekaterinburg, Russian Federation

3 Kostanay Oblast Tuberculosis Dispensary, Kostanay, Republic of Kazakhstan 

4 Department of Medical Biology and Genetics, Faculty of Medicine, Demiroglu Bilim University, Istanbul, Turkey

 5 Diabetes Application and Research Center, Demiroglu Bilim University, Istanbul, Turkey

6 Department of Immunology, Aziz Sancar Institute of Experimental Medicine, Istanbul University, Istanbul, Turkey

7 Diabetes Application and Research Center, Istanbul University, Istanbul, Turkey

8 International Diabetes Center, Acibadem University, Istanbul, Turkey

2) We remove any funding-related text from the manuscript. 

We expand the acronym “IIP UB RAS”.

In the Funding Statement section of the online submission form one’s should to note the follow:

“This research was carried out within the state assignment of the Institute Immunology and Physiology Ural Branch Russian Academy of Science (IIP UB RAS) № 122020900136-4. The funders had no role in study design, data collection and analysis, decision to publish, or preparation of the manuscript. The work was performed using the equipment of the Shared Research Center of Scientific Equipment of the IIP UB RAS.”

---

## [Decision Letter · Decision Letter 1]

5 Oct 2023

PONE-D-23-22087R1Hepatic Insulin Synthesis Increases in Rat Models of Diabetes Mellitus Type 1 and 2 DifferentlyPLOS ONE

Dear Dr. Aydin Ozgur,

Thank you for submitting your manuscript to PLOS ONE. After careful consideration, we feel that it has merit but does not fully meet PLOS ONE’s publication criteria as it currently stands. Therefore, we invite you to submit a revised version of the manuscript that addresses the points raised during the review process.

We look forward to receiving your revised manuscript.

Kind regards,

Riham M. Aly

Academic Editor

PLOS ONE

Journal Requirements:

Reviewers' comments:

Reviewer's Responses to Questions

**Comments to the Author**

1. If the authors have adequately addressed your comments raised in a previous round of review and you feel that this manuscript is now acceptable for publication, you may indicate that here to bypass the “Comments to the Author” section, enter your conflict of interest statement in the “Confidential to Editor” section, and submit your "Accept" recommendation.

Reviewer #1: All comments have been addressed

2. Is the manuscript technically sound, and do the data support the conclusions?

Reviewer #1: Yes

3. Has the statistical analysis been performed appropriately and rigorously? 

Reviewer #1: Yes

4. Have the authors made all data underlying the findings in their manuscript fully available?

Reviewer #1: Yes

5. Is the manuscript presented in an intelligible fashion and written in standard English?

Reviewer #1: Yes

6. Review Comments to the Author

Reviewer #1: Reviewer Recommendations

I would like to thank the authors of manuscript entitled "Hepatic Insulin Synthesis Increases in Rat Models of Diabetes Mellitus Type 1 and 2 Differently” for their effort in making the suggested modifications, Now the manuscript showed great improvement, yet some minor modifications are recommended.

Introduction:

Add reference to this paragraph (Extra-islet and extra-pancreatic IPCs are the spotlight of researchers for at least two reasons: first, they are additional sources of insulin and probably are able to alleviate hyperglycemia and prevent complications of diabetes. Secondly, ectopic insulin expression in functional mature cell in different tissues is of fundamental theoretical interest and expand the understanding of tissue adaptation in pathology condition (Reference???).

RESULTS

Remove number 38 from this paragraph: (Rats with T2D showed poor glucose tolerance, as evident by higher AUC value observed in T2D group compared to ND group (p < 0.05) (Fig. 1С) [38]).

Figure 2 & 5 legends: Cite the group according to figure letters (A, B &C):

(A) ND – non-diabetic group, (B) T1D – type 1 diabetes, (C) T2D – type 2 diabetes.

Reorganize the text to be matching with figure order:

Markers of liver injury and inflammation in blood

To evaluate functional state of hepatocytes in the models of T1D and T2D, we analyzed

activity of aminotransferases (AST and ALT) instead of (ALT and AST), alkaline phosphatase (ALP) instead of (AP) and protein content in blood plasma (Fig. 3A-E). There was an increase in serum AST level in T1D, compared to ND. In contrast, AST level in T2D did not change. ALT is a rather specific marker of hepatocellular injury [45]. Increase in ALT level compared to normal was higher in T2D group than in T1D. A high ALT level led to a decrease in AST/ALT ratio (De Ritis coefficient) in T2D. ((((There was an increase in serum AST level in T1D, compared to ND. In contrast, AST level in T2D did not change)))) REMOVE the sentence.

Add letter (E) to figure 3

7. PLOS authors have the option to publish the peer review history of their article (what does this mean?). If published, this will include your full peer review and any attached files.

Reviewer #1: No

---

## [Author Response · Author response to Decision Letter 1]

31 Oct 2023

Dear colleagues, 

We would like to thank you for your helpful comments. Please finf below a point-by-point response to your comments and questions.

Response to Reviewer #1 Comments:

Point 1: 

Introduction:

Add reference to this paragraph -

«Extra-islet and extra-pancreatic IPCs are the spotlight of researchers for at least two reasons: first, they are additional sources of insulin and probably are able to alleviate hyperglycemia and prevent complications of diabetes. Secondly, ectopic insulin expression in functional mature cell in different tissues is of fundamental theoretical interest and expand the understanding of tissue adaptation in pathology condition» (Reference???).

Response 1: 

Done.

Point 2: 

RESULTS

Remove number 38 from this paragraph: 

«Rats with T2D showed poor glucose tolerance, as evident by higher AUC value observed in T2D group compared to ND group (p < 0.05) (Fig. 1С) [38])».

Response 2:

Done.

Removed reference was the article of Sakaguchi, K. et al. Glucose area under the curve during oral glucose tolerance test as an index of glucose intolerance. Diabetol Int 2016, 7(1), 53–58, which discuss the usage of an oral glucose tolerance test (OGTT) for clear detection of glucose intolerance in the early stage of diabetes.

Point 3: 

Figure 2 & 5 legends: Cite the group according to figure letters (A, B &C):

(A) ND – non-diabetic group, (B) T1D – type 1 diabetes, (C) T2D – type 2 diabetes.

Response 3:

Done.

Point 4: 

Reorganize the text to be matching with figure order:

Markers of liver injury and inflammation in blood

To evaluate functional state of hepatocytes in the models of T1D and T2D, we analyzed

activity of aminotransferases (AST and ALT) instead of (ALT and AST), alkaline phosphatase (ALP) instead of (AP) and protein content in blood plasma (Fig. 3A-E). There was an increase in serum AST level in T1D, compared to ND. In contrast, AST level in T2D did not change. ALT is a rather specific marker of hepatocellular injury [45]. Increase in ALT level compared to normal was higher in T2D group than in T1D. A high ALT level led to a decrease in AST/ALT ratio (De Ritis coefficient) in T2D. ((((There was an increase in serum AST level in T1D, compared to ND. In contrast, AST level in T2D did not change)))) REMOVE the sentence.

Response 4:

Done.

Point 5: Add letter (E) to figure 3

Response 5:

Done.

---

## [Editor Report · Decision Letter 2]

2 Nov 2023

Hepatic Insulin Synthesis Increases in Rat Models of Diabetes Mellitus Type 1 and 2 Differently

PONE-D-23-22087R2

Dear Dr. Aydin Ozgur,

We’re pleased to inform you that your manuscript has been judged scientifically suitable for publication and will be formally accepted for publication once it meets all outstanding technical requirements.

Kind regards,

Dr Riham M. Aly

Academic Editor

PLOS ONE
---

## [Editor Report · Acceptance letter]

17 Nov 2023

PONE-D-23-22087R2 

Hepatic Insulin Synthesis Increases in Rat Models of Diabetes Mellitus Type 1 and 2 Differently 

Dear Dr. Aydin Ozgur:

I'm pleased to inform you that your manuscript has been deemed suitable for publication in PLOS ONE. Congratulations! Your manuscript is now with our production department. 

Kind regards, 

on behalf of

Dr. Riham M. Aly 

Academic Editor

PLOS ONE